# Fast growth and high-titer bioproduction from renewable formate via metal-dependent formate dehydrogenase in *Escherichia coli*

Aidan E. Cowan[1,2] ✉, Mason Hillers[1], Vittorio Rainaldi [3], Florent Collas[4], Hemant Choudhary [1,5], Basem S. Zakaria[6,7], Gregory G. Bieberach [4], David N. Carruthers[1,7], Maxwell Grabovac[1], Jennifer W. Gin[1], Bridgie Cawthon[1], Yan Chen[1], Emine Akyuz Turumtay[1], Edward E. K. Baidoo[1], Christopher J. Petzold [1], Adam M. Feist[1,8,9], Sara Tejedor-Sanz[6,7], Frank Kensy[4], Blake A. Simmons [1,7], Jay D. Keasling [1,7,9,10,11] ✉ & Nico J. Claassens [3] ✉

Microbial bioproduction using one-carbon (C1) feedstocks has the potential to decarbonize the manufacturing of materials, fuels, and chemicals. Formate is a promising C1 feedstock, and the realization of industrial, formatotrophic platform organisms is a key goal for C1-based bioproduction. So far, a major limitation for synthetic formatotrophy has been slow energy supply due to slow formate dehydrogenase activity. Here, we implement a fast, metal-dependent formate dehydrogenase complex in a synthetic formatotrophic *Escherichia coli* utilizing the reductive glycine pathway. After a short-term evolution, we demonstrate formatotrophic growth of *E. coli* with a doubling time of less than 4.5 h, comparable to the fastest natural formatotrophs. To further explore the potential of a formate-based bioeconomy, this strain is engineered to produce mevalonate, as well as the terpenoid and aviation fuel precursor isoprenol, using formate we generate directly from the electrochemical reduction of $CO_2$. This work demonstrates an improvement in bioproduct titer from formate, achieving the production of 3.8 g/L of mevalonate. Additionally, the abundant and recalcitrant polymer lignin is chemically decomposed into a formate-rich mixture of small organic acids and subsequently bioconverted into mevalonate. Overall, the described fast-growing, formatotrophic bioproduction strain demonstrates that a sustainable formate bioeconomy is within reach.

Anthropogenic greenhouse gas emissions are disrupting the climate systems in which modern society has developed. These perturbations are endangering biodiversity, shifting the distribution of habitable land, and limiting economic growth with potentially devastating consequences for vulnerable ecosystems and communities[1–3]. Alternatives to petrochemicals and fossil fuels are urgently needed to avoid the most ruinous effects of climate change.

State-of-the-art microbial bioproduction, where a sugar feedstock is converted to a useful product by a microorganism, has the potential to decrease the carbon intensity of chemical industries[4,5]. However, the

use of sugar as a feedstock at scale necessitates large land use changes and creates competition with land used for the cultivation of food. Additionally, the conversion of wildland to farmland threatens biodiversity and causes the release of greenhouse gases either from agricultural practices such as tilling and fertilizer use or by the disruption of the natural carbon cycles[6,7].

As an alternative to sugar feedstocks, one-carbon (C1) feedstocks have the potential to further decrease the carbon footprint of microbial bioproduction without using agricultural land. C1 feedstocks can be generated electrochemically from $CO_2$ using renewable energy in a manner that is both highly energy-efficient and carbon-negative[8-10]. Alternatively, some C1 feedstocks can also be obtained from organic waste streams[11-14]. Among potential C1 feedstocks, formate has advantages such as low toxicity to humans and is fully soluble in liquid culture[15]. However, to realize a formate bioeconomy, where formate serves as a mediator molecule between electrocatalytic and biological systems, suitable industrial formatotrophic hosts are needed. Most natural formatotrophs lack extensive genetic toolboxes and metabolic knowledge, which complicates the development of bioproduction strains and limits the titers achieved in these hosts to date[16,17]. Alternatively, biotechnological platform organisms, such as *Escherichia coli* and *Saccharomyces cerevisiae*, which natively cannot grow on formate, can be engineered to utilize formate as a carbon source[18-21].

In nature, there are three dominant metabolic pathways supporting growth on formate: the Calvin cycle, the serine cycle, and the Wood-Ljungdahl pathway[22]. The first two pathways have relatively high ATP costs, which leads to lower biomass and product yields[23,24]. The Wood-Ljungdahl pathway has served as a highly efficient and successful pathway for bioproduction of small molecules such as ethanol[25]. However, the Wood-Ljungdahl pathway relies upon highly oxygen-sensitive enzymes and cannot regenerate additional ATP through oxidative phosphorylation with $O_2$ as an electron acceptor. Hence, the Wood-Ljungdahl pathway cannot be used to efficiently produce chemicals for which biosynthesis routes require an input of ATP[26]. For example, this pathway cannot efficiently synthesize the valuable class of ATP-dependent terpenoid products, with diverse applications in fuels, plastics, flavors, and fragrances[27].

As an alternative to the natural routes, the reductive glycine pathway (rGlyP) was considered as a highly promising synthetic route for ATP-efficient, aerobic formate assimilation[22]. Several studies have engineered the rGlyP, culminating in the realization of full formatotrophic growth of *E. coli* and *Cupriavidus necator*[18,19,28-34]. Partial formate assimilation via the rGlyP has been demonstrated in the bioproduction hosts *S. cerevisiae* and *Pseudomonas putida*[20,35,36].

Despite these major breakthroughs, none of these studies has realized comparable growth rates to the fastest natural formatotrophs, which have doubling times of ~4 h[23,37,38]. The fastest synthetic formatotroph has a doubling time longer than 6 h, and production of lactate and polyhydroxybutyrate (PHB) by this *E. coli* strain was limited[33,39]. This illustrates the need for fast and efficient bioproduction platform strains.

A potential kinetic bottleneck preventing fast, synthetic formatotrophic growth is the heterologous formate dehydrogenase (FDH) enzyme, which generates the NADH necessary to reduce formate to metabolic intermediates and biomass[22]. All formatotrophic *E. coli* strains constructed to date utilize a simple, single-subunit, metal-independent FDH from *Pseudomonas* sp. 101 (psFDH)[18,19,33,40]. However, this enzyme has relatively slow turnover ($k_{cat} \approx 10 \text{ s}^{-1}$) compared to the more complex, metal-dependent FDHs ($k_{cat} > 100 \text{ s}^{-1}$)[41,42]. Metal-dependent FDHs are a diverse class of enzymes that reversibly catalyze the oxidation of formate at an active site containing molybdenum or tungsten. Electrons from formate flow through a series of iron-sulfur clusters to a distal active site where an electron carrier, such as $NAD^+$, is reduced. The most suitable metal-dependent replacements for psFDH are the class 5 FDHs of which *C. necator* (cnFDH) and

*Rhodobacter capsulatus* (rcFDH) are model enzymes[43,44]. Crucially, these enzymes are uniquely oxygen tolerant and $NAD^+$-dependent.

In this work, we replace the metal-independent psFDH with a metal-dependent cnFDH in a formatotrophic *E. coli* strain equipped with the rGlyP. A brief adaptive laboratory evolution campaign yields a strain with a doubling time of $\approx 4.5$ h, comparable to the fastest growing natural formatotrophs. We employ the resulting formatotrophic strain for the production of mevalonate, a precursor to terpenoids and bioplastics, achieving a titer of 3.8 g/L mevalonate from formate alone. As a result, we establish synthetic formatotrophs as efficient bioproduction chassis and further demonstrate their integration into platforms where $CO_2$ can be fixed electrolytically as e-formate. To address potential economic limitations to the widespread use of e-formate[9,45,46], we also demonstrate the oxidation of lignin, an abundant and underutilized waste product[47,48], to a formate-rich mixture of organic acids with subsequent bioconversion to mevalonate by the generated formatotrophic platform strain.

## Results

### Demonstrating functional expression of a metal-dependent formate dehydrogenase in an energy-auxotrophic E. coli

As an intermediate step towards the reconstitution of cnFDH in a fully formatotrophic *E. coli*, cnFDH was reconstituted in an energy-auxotrophic strain of *E. coli* (Fig. 1)[49]. This strain contains a knockout of the *lpd* gene, which encodes lipoamide dehydrogenase, an essential subunit of the alpha-ketoglutarate and pyruvate dehydrogenase complexes. Without the activity of these enzyme complexes, *E. coli* cannot generate sufficient NADH to sustain growth on acetate as a sole carbon source. However, growth can be restored by expressing a heterologous NADH-regenerating system (Fig. 1b). Therefore, this strain serves as a sensor for NADH regenerating activity within the cell and allows for testing cnFDH in vivo.

The genes comprising the cnFDH operon, *fdsGBACD*, along with their native RBSs, were amplified from the genomic DNA of *C. necator* and inserted into a low copy, IPTG-inducible BglBrick *E. coli* expression vector, resulting in pBbS1k_cnFDH[50]. Next, as a control plasmid, psFDH was inserted into the same backbone to yield pBbS1k_psFDH. Transformation of *E. coli* Δ*lpd* with pBbS1k_cnFDH and pBbS1k_psFDH created strains Δ*lpd*_cnFDH and Δ*lpd*_psFDH; the growth of these strains was measured to confirm that cnFDH was functionally expressed and to compare the in vivo activity of each FDH. Both FDHs allowed for the growth of the energy auxotroph when the plasmids were induced with IPTG and formate was present (Fig. 1c, Supplementary Fig. 1). No growth was observed when formate was absent or when the strain lacked a heterologous FDH. The minimum doubling time of the strain containing pBbS1k_cnFDH was significantly lower compared to pBbS1k_psFDH (Fig. 1d). The relative magnitude of the growth rates is consistent with the relative magnitude of reported $k_{cat}$ values for the two enzymes, with cnFDH having an order of magnitude higher $k_{cat}$[41,42]. This result was achieved with cnFDH occupying only 0.2% of the proteome of Δ*lpd*_cnFDH compared to greater than 20% proteome allocation to psFDH in strain Δ*lpd*_psFDH (Fig. 1e). These results indicate that the complex process of molybdoenzyme maturation in *E. coli* was sufficient for heterologous expression of cnFDH in a functional state. Furthermore, these data indicate that cnFDH is a more efficient energy conversion system than psFDH, enabling faster growth of our sensor strain with far less proteome allocation.

### Introducing a metal-dependent FDH and evolving E. coli towards fast formatotrophic growth

To demonstrate the utility of a kinetically faster and more efficient formate dehydrogenase in a more biotechnologically relevant context, cnFDH was leveraged to generate reducing equivalents (NADH) required for formatotrophic growth via the rGlyP in *E. coli*. Given that formate is relatively oxidized compared to biomass, the rGlyP requires

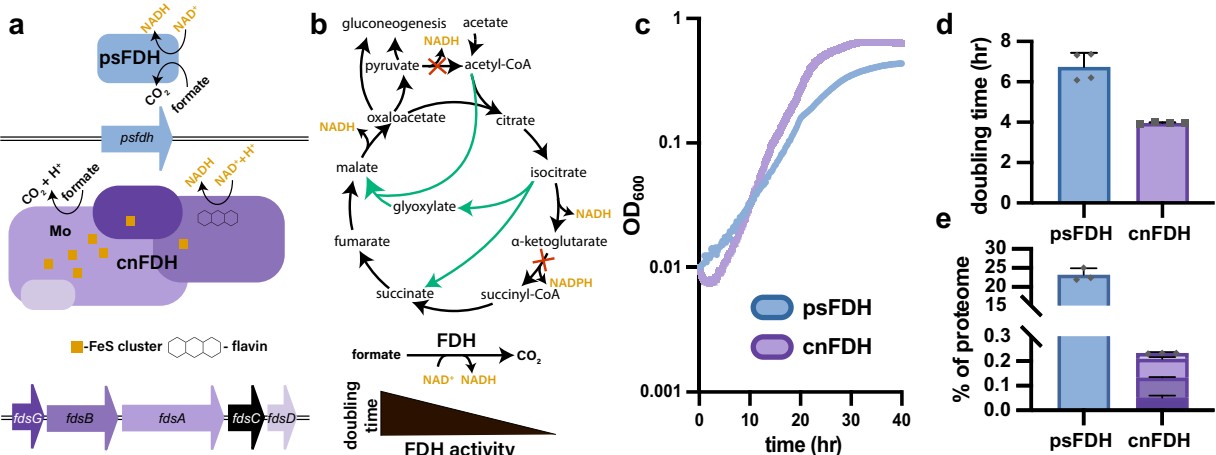

**Fig. 1 | Functional reconstitution of a metal-dependent formate dehydrogenase from *C. necator* in *E. coli*. a** Schematic diagram of FDH proteins and genes used in this study highlighting the structural complexity of cnFDH and cofactors required for functionality. **b** Schematic diagram of the metabolism of *E. coli* Δ*lpd* showing reactions disrupted by the deletion of *lpd* (red cross) and the flux of carbon through the non-oxidative glyoxylate shunt (green). **c** Growth curves showing the rescue of the Δ*lpd* strain in a minimal medium containing 20 mM acetate and 60 mM formate (*n* = 4). Only the induction level that minimized the doubling time for each condition (0.5 mM IPTG for pBbS1k_psFDH and 0.05 mM IPTG for pBbS1k_cnFDH) is shown here for clarity (additional induction conditions are shown in

Supplementary Fig. 1). **d** Extracted doubling times from the growth curves displayed in c showing a significant (*p* = 0.004) decrease in doubling time of Δ*lpd* with rescue by cnFDH (*n* = 4). Statistical analysis was performed using a two-tailed *t*-test. **e** Proteome allocation to FDH protein subunits in each condition shown in c and d. psFDH accounted for roughly 100 times more of the proteome than the cnFDH complex, likely imparting a burden which could explain the difference in doubling time (*n* = 4). Individual data points are shown as the sum of all FDH subunits within each biological sample. Data represent the mean ± s.d. of biological replicates. Source data are provided as a Source Data file.

the activity of an energy module to generate NADH. This NADH can then be used to reduce formate, enabling its conversion into biomass and product precursors such as pyruvate (Fig. 2a). To accomplish this, the original psFDH energy module present in the genome of the formatotrophic rGlyP strain K4e[18] was deleted, and the resulting strain (K4e *psFDH::cat*) was unable to grow solely on formate. Introduction of pBbS1k_cnFDH enabled growth of K4e *psFDH::cat* with a prolonged lag phase (Supplementary Fig. 2a), and this strain is further referred to as K4M (molybdenum-dependent FDH). Surprisingly, growth was also observed without induction of the cnFDH operon with IPTG (Supplementary Fig. 2a). To fine-tune expression of cnFDH to support faster growth, we chose to use adaptive laboratory evolution (ALE)[51]. We expect that ALE is the best method to enable rapid fine-tuning of cnFDH levels to optimal levels for fast growth. Tuning of FDH expression through short-term ALE was also necessary to achieve faster growth of the original K4e strain, and is therefore important to compare these two strains[18]. We selected for fast formatrophic growth by ALE in the absence of IPTG, to enable formatotrophic growth without the need to add an inducer to the culture medium. The ALE consisted of propagating K4M in a minimal medium containing 60 mM formate for approximately 25 generations in one replicate evolution.

Three clones of the adaptively evolved culture were isolated from the single ALE experiment and their genomes as well as the pBbS1k_cnFDH plasmid were sequenced (Fig. 2b). Each evolved clone possessed the same mutation in the lac operator of pBbS1k, which tuned the expression of cnFDH (Fig. 2e) to a more optimal level, by de-repressing expression in the absence of IPTG and increasing its expression overall, as later determined by proteomic analysis. The initial growth of K4M before FDH expression tuning was likely severely hindered by low levels of this key enzyme, which is required for energy generation (Supplementary Fig. 2a). Hence, single mutations that allow for the appropriate level of expression of the FDH enzyme lead to a large increase in growth rate. This explains why this mutation tuning the expression of the cnFDH promoter emerged rapidly over a short ALE campaign. Whole genome sequencing also revealed only two distinct genomic mutations in the evolved clones, both in the gene *rpoC* (Fig. 2b). The exact effect of these SNPs is unclear. *rpoC* encodes

the β′ subunit of *E. coli* RNA-polymerase and hence mutations in this gene likely impact global transcription. Different mutations in *rpoC* have been shown to improve the growth of K4e and of *E. coli* in general in minimal media[33,52]. The three evolved strains were named K4Me1-3. K4Me2 and 3 were genetically identical, whereas K4Me1 contained a distinct *rpoC* mutation. The mutations associated with a given strain are shown visually in Fig. 2b and in tabular form in Supplementary Table 1.

To isolate the effect of the plasmid mutation from the effect of the *rpoC* mutations, the mutated pBbS1k_cnFDH plasmid (pBbS1k_cnFDH*) was purified and used to retransform the original K4e *psfdh::cat* strain (generating K4M*). The doubling times of K4Me1, K4Me2, K4M*, and the original K4e were determined. The three K4M-based strains all grew faster than the original K4e strain and reached their final OD in less time (Fig. 2c, d). Overall, K4Me2 grew the fastest with a minimum doubling time of 4.4 ± 0.1 hours (Fig. 2d). This doubling time represents a significant improvement over the growth of previously reported synthetic formatotrophs. Moreover, this rate approaches the fastest reported formatotrophic doubling time via the Calvin cycle of 3.3 h, which was recently achieved for an evolved *C. necator* strain after long-term ALE on formate (Supplementary Table 2)[38]. The biomass yield was also improved in K4Me2 (3.3 ± 0.1 gDCW/mol formate) relative to K4e (2.3 ± 0.1 gDCW/mol formate) (Supplementary Fig. 2b). This enhancement in growth characteristics was achieved through a relatively short ALE (25 vs. 200 generations) compared to a longer ALE conducted on the original K4e strain, which produced the K4e2 strain with a doubling time of 6 h[33]. In summary, the replacement of the FDH and optimization via ALE suggested that the growth of the original K4e strain was limited by psFDH and that a rational intervention followed by ALE was able to achieve a lower doubling time inaccessible to the original strain. Despite a key mutation (lacO nt. 8 G to A) in the evolved strains, which increased the expression of cnFDH from its untuned level in K4M (Fig. 2e), the overall expression of the cnFDH complex is still twenty-fold lower than the expression of psFDH in the K4e strain (Supplementary Fig. 2c), imparting less expression burden and potentially explaining the decreased doubling time. These results bear strong implications for the utility of K4Me strains as industrial hosts

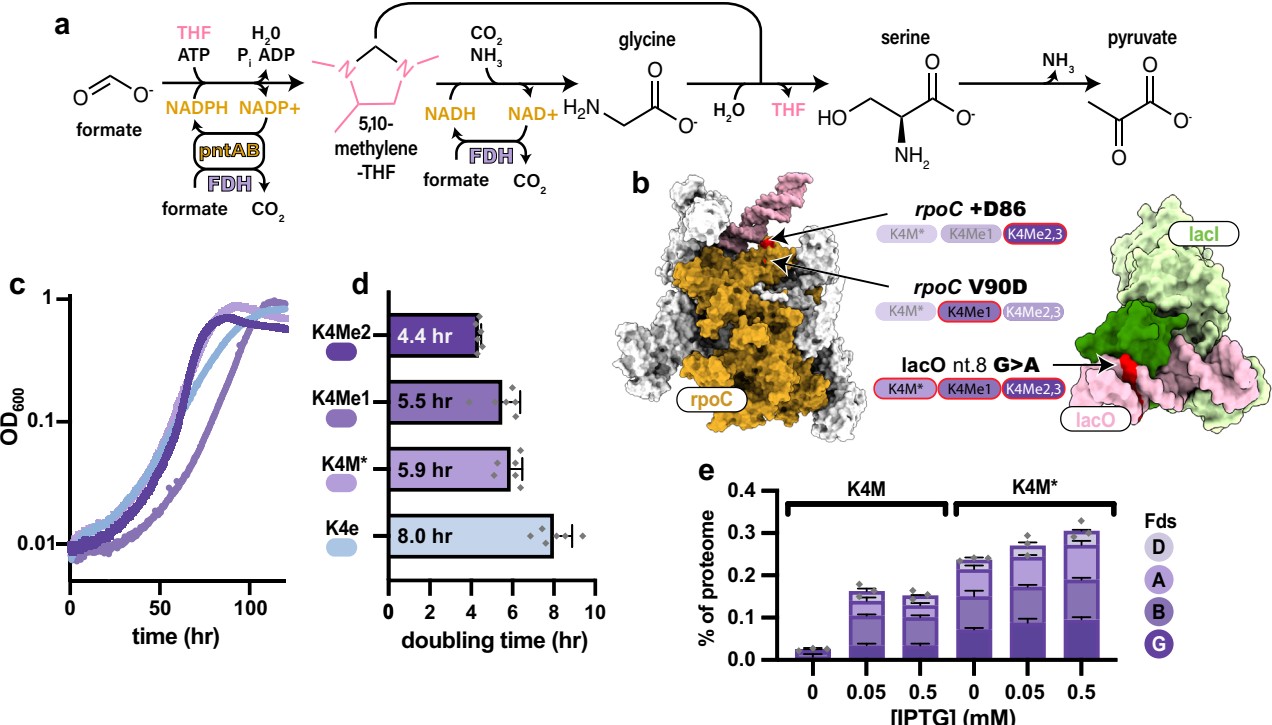

**Fig. 2 | Replacement of the psFDH with the cnFDH to support faster growth via the rGlyP. a** Schematic showing reactions and selected intermediates in the rGlyP and cofactors required for its activity. b, Protein/DNA complexes acquiring mutations during a short term adaptive laboratory evolution are shown (left: *E. coli* RNA polymerase (orange: RpoC, pink: upstream fork promoter DNA) PDB:6N62; Right: LacI dimer bound to lac operator (green: LacI dimer, pink:LacO DNA) PDB:1EFI). Mutated amino acids/nucleotides in each complex are colored red and labeled. Strains containing the indicated mutation are denoted by a red circle around their respective label. **c** Representative growth curves of the K4e and K4M strains in 100 mM formate minimal media. **d** Doubling times from (**c**). All differences

between K4e and K4M strains are significant (K4M* $p = 0.0007$, K4Me1 $p = 0.0007$, K4Me2 $p = 0.0001$) ($n = 6$). Statistical analysis was performed using a two-tailed $t$-test. **e** Proteomic analysis of cnFDH expression in the K4M base strain compared to K4M* isolating the effect of the lacO mutation. Proteomics revealed a decrease in repression due to a mutation in the lac operator as well as overall higher levels of expression. Individual data points are shown as the sum of all FDH subunits within each biological sample. For the proteomic analysis, strains were cultivated in M9 with 5 g/L yeast extract and 2% glucose to avoid a selective pressure imparted by growth on formate, which could modify expression level ($n = 3$). Data represent the mean ± s.d. of biological replicates. Source data are provided as a Source Data file.

and the utility of heterologous cnFDH in engineered microbial systems.

## Bioproduction of mevalonate by the metal-dependent FDH strain achieves high titers

To determine the suitability of the K4M-derived strains as bioproduction chassis, we sought to demonstrate the production of an industrially relevant molecule from formate. Given that K4M's primary modification was in a redox generating pathway, we chose to demonstrate the production of a reduced product. Mevalonate was chosen given its commercial applications as a bioplastic monomer[53]. Mevalonate is also the precursor to isoprenoids, a diverse group of chemicals with many uses[5,54]. In the conventional pathway, mevalonate is formed through the condensation of three acetyl-CoA units followed by a four-electron reduction (Fig. 3a).

To enable mevalonate bioproduction, K4e and the K4M-derived strains were transformed with the pBbA5a_mevalonate plasmid for expression of acetyl-CoA acetyltransferase (atoB), HMG-CoA synthase (HMGS), and HMG-CoA reductase (HMGR) creating strains K4e_mev, K4M*_mev, K4Me1_mev, and K4Me2_mev. For the purposes of initial characterization of relative differences in product titers between stains, bioproduction was first conducted at a small (3 mL) scale in mixotrophic batch mode with 100 mM formate and 1 mM glucose. The addition of glucose facilitated the growth of the strains in these non-optimal, small-scale batch conditions, to allow for the determination of the best performing strain, which could then be used for bioproduction

from formate alone at larger scale in fed-batch. All strains produced mevalonate, with the K4M-derived strains each reaching a significantly higher final titer. K4e_mev reached a maximum mevalonate titer of only 37.5 ± 8.8 mg/L, whereas K4M*_mev, K4Me1_mev, and K4Me2_mev produced 67.7 ± 9.1 mg/L, 74.5 ± 2.9 mg/L, and 86.8 ± 7.3 mg/L, respectively (Fig. 3b). No mevalonate was detected in strains lacking pBbA5a_mevalonate.

Interestingly, the strains' maximum titer was inversely correlated with their respective final $OD_{600}$ (Supplementary Fig. 3b). This indicated that the metabolic flux was diverted away from biomass formation and towards bioproduct formation to a greater extent in the K4M-derived strains. To determine the mechanism by which the K4M-derived strains were able to achieve higher titers, we used proteomic analysis to determine the expression of the mevalonate pathway genes in all of these strains during bioproduction. This analysis revealed that the K4M-derived strains devoted a larger percentage of their proteome to the mevalonate pathway enzymes (up to ≈8% more) (Fig. 3c). The use of cnFDH as an energy generation system may have allowed for reallocation of the ≈4% of the proteome (Supplementary Fig. 2c), which was previously dedicated to psFDH, to mevalonate pathway expression. Increased mevalonate pathway expression would explain the increased flux into mevalonate observed in the strains expressing cnFDH. Interestingly, the final titers achieved for each strain were highly correlated ($R^2 = 0.947$) with the expression level of AtoB (Supplementary Fig. 3c), the first dedicated step in the mevalonate pathway. It stands to reason that the expression level of AtoB may dictate

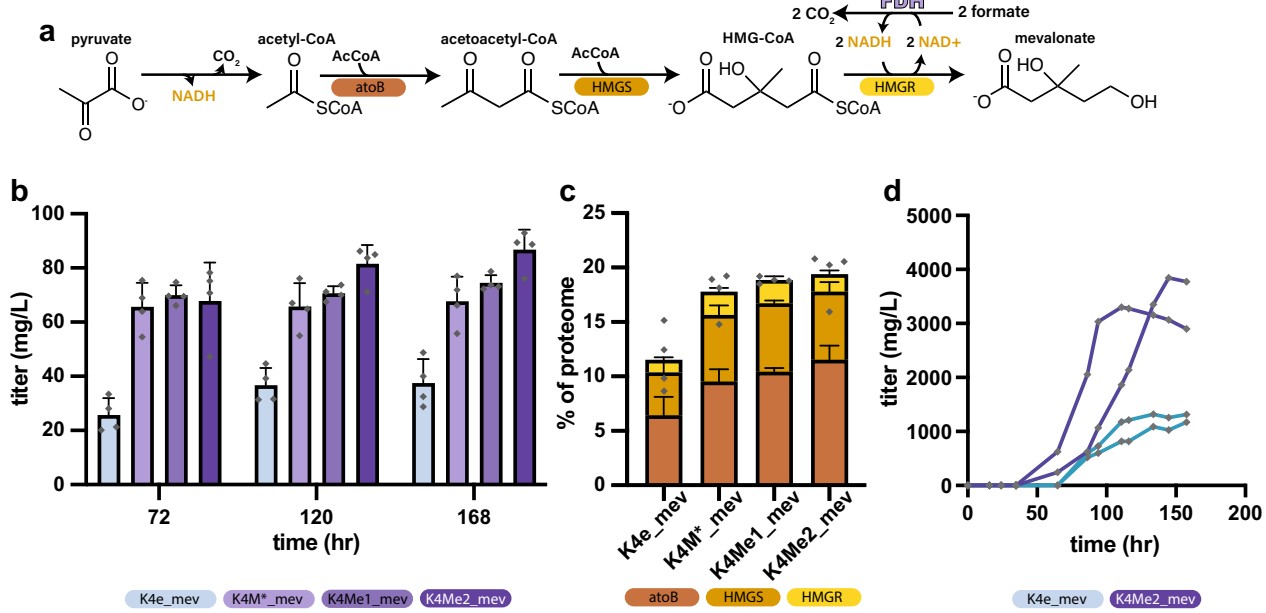

**Fig. 3 | Bioproduction of mevalonate from formate. a** Schematic depicting the mevalonate pathway by which the output metabolite of the rGlyP, pyruvate (left), is transformed into mevalonate (right). Heterologously expressed enzymes catalyzing these reactions are denoted below their respective reactions. **b** Mixotrophic bioproduction of mevalonate conducted with 100 mM formate and 1 mM glucose to assay differences in bioproduction at small scale in batch mode. All differences between K4e and K4M strains are significant (K4M*_mev $p$ = 0.003, K4Me1_mev $p$ = 0.002, K4Me2_mev $p$ = 0.0002) ($n$ = 4). **c** Analysis of the proteomic allocation to mevalonate biosynthetic enzymes. The expression of each gene in all K4M strains is significantly higher than in K4e (K4M*_mev $p$ = 0.01, K4Me1_mev $p$ = 0.002,

K4Me2_mev $p$ = 0.005) ($n$ = 4). Individual data points are shown as the sum of the mevalonate pathway enzymes within each biological sample. **d** Fed-batch bioproduction using K4e and the most productive strain from small-scale, batch experimentation (K4Me2). All strains were induced with 0.5 mM IPTG at 65 h post inoculation to induce bioproduction. K4Me2 produced a maximum titer of 3.3 and 3.8 g/L while K4e produced significantly less mevalonate ($p$ = 0.02) with a maximum titer of 1.3 and 1.2 g/L ($n$ = 2). Data represent the mean ± s.d. of biological replicates. All statistical analysis was performed using a two-tailed $t$-test. Source data are provided as a Source Data file.

the metabolic flux that is diverted from biomass accumulation to mevalonate and that increased expression could increase flux into the mevalonate pathway. Final titer was also correlated with doubling time, though doubling time, which is ultimately dependent on the rate of formate assimilation, would be most expected to correlate with productivity (which was also increased in K4Me2 relative to K4e (Supplementary Fig. 4c)).

While small-scale, batch bioproduction is useful due to throughput and ability to screen multiple strains it poorly approximates conditions encountered in industrial bioproduction. We assayed our best performing strain, K4Me2_mev, at a larger scale in fed-batch to determine the titer this strain could produce. For comparison, we also tested strain K4e_mev in the larger-scale fed-batch setup. Batch bioproduction using formate significantly limits the amount of formate feedstock that can be used (and therefore the final titer), as *E. coli* (like natural formatotrophs such as *C. necator*) cannot tolerate formate concentrations above 100-200 mM[18,55]. However, in fed-batch, higher amounts of formate can be fed as formate consumed by the host can be replaced by formic acid to maintain a constant, non-toxic concentration of formate and a constant pH. Under fed-batch conditions with formate as the sole carbon source, the two K4Me2 cultures reached a maximum titer of 3.3 and 3.8 g/L, whereas K4e only achieved titers of 1.3 and 1.2 g/L (Fig. 3d). The maximum titers for the K4Me2 bioproduction are compared to previously reported attempts at fully formatotrophic bioproduction in Supplementary Table 3[55]. In agreement with the small-scale results, K4Me2 grew to a lower final $OD_{600}$ (Supplementary Fig. 4a), despite consuming similar or slightly greater amounts of formate (Supplementary Fig. 4b), again indicating more substantial diversion of metabolic flux in K4Me2 where carbon flows into the product at the

expense of final OD. A trade-off between growth and product titer is a well-recognized characteristic of many bioproduction chassis. However, optimizing for production at the expense of growth can be difficult to engineer in microbial systems which are under constant selective pressure for improved growth.

## Demonstration of bioproduction of mevalonate and isoprenol from e-formate

The promise and utility of this work are to advance the possibility of a carbon-neutral or carbon-negative formate bioeconomy, where $CO_2$-derived formate serves as a supply of carbon and energy for microbial cell factories to produce fuels, materials, and chemicals. To demonstrate this possibility, we sought to generate and utilize streams of $CO_2$-derived formate using K4Me2. The following sections seek to demonstrate the utility of these engineered formatotrophic bioproduction chassis within the context of a formate bioeconomy.

Formate can be efficiently generated through electrochemical means by the reduction of $CO_2$[9,45,46]. To demonstrate this process, we used a small-scale, dual-chambered electrolyzer to reduce $CO_2$ and carbonate to formate in a bio-compatible potassium carbonate catholyte solution. The resulting catholyte solution contained 1.3 M of electrochemically-derived formate (e-formate) (Fig. 4a), and no other products were detected by HPLC-RID besides formate. The catholyte solution was diluted into M9 medium to a final concentration of 100 mM. This medium containing $CO_2$-derived formate was then used to culture K4Me2_mev in small-scale batch mode (Fig. 4b), yielding mevalonate at a final titer of 22.8 ± 3.0 mg/L (Fig. 4c). The catholyte solution showed no significant inhibitory effects on growth (Fig. 4b) and did not necessitate purification before use, simplifying its use as a carbon and energy source.

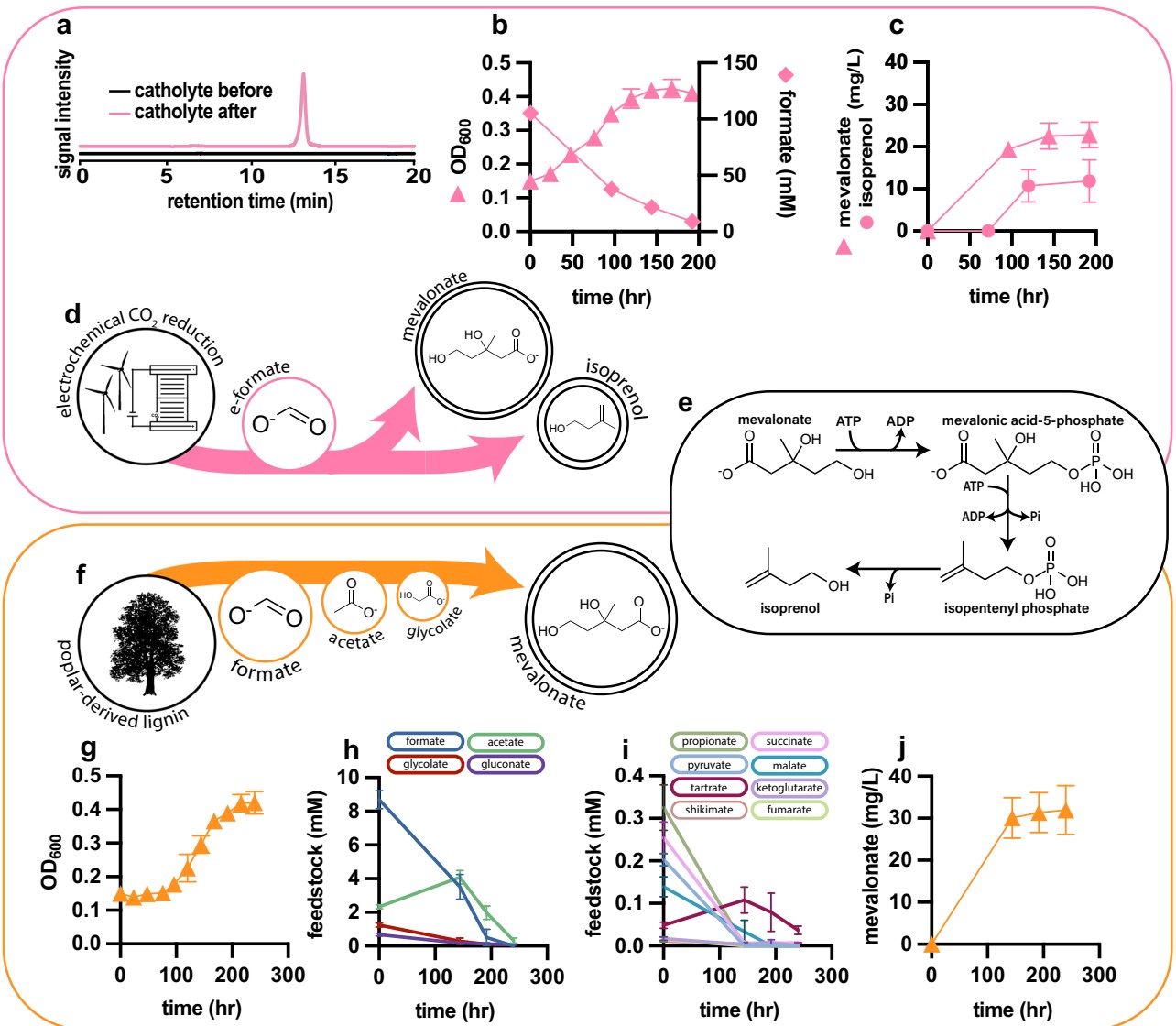

**Fig. 4 | Renewable bioproduction from both CO₂-derived e-formate and a lignin-derived organic acid stream containing formate. a** Electrochemical production of formate from CO₂. Catholyte solution was analyzed by HPLC-RID before and after electrochemical formate production and shows only one single peak corresponding to formate (chromatograms are offset for clarity). **b** Consumption of 100 mM formate during the bioproduction of mevalonate from electrochemically derived formate and optical density over the course of cultivation. Formate consumption mirrors cell growth during formatotrophic bioproduction of mevalonate ($n = 4$). **c** Mevalonate and isoprenol titers during two parallel bioproduction experiments from electrochemically produced formate; the maximum titers achieved were 22.9 (±3.0) mg/L mevalonate and 11.9 (±5.6) mg/L isoprenol. Mevalonate was produced from 100 mM electrochemically produced formate ($n = 4$). Isoprenol was produced from 120 mM electrochemically produced formate provided as a 60 mM initial concentration followed by another 60 mM addition at 72 h ($n = 3$). **d** Schematic diagram of mevalonate and isoprenol production from electrochemically reduced CO₂. Bioproducts are shown below in circles whose area correlates to the titer achieved at small scale (positive error denoted by larger concentric circle). **e** IPP bypass pathway from mevalonate to isoprenol. **f** Schematic diagram of mevalonate bioproduction from lignin-rich acid mixture. CO₂ is fixed into lignin through photosynthesis. Lignin is oxidatively depolymerized to formate and other small acids before bioproduction. **g** OD₆₀₀ of K4Me2 growing on formate-containing acid mixture derived from lignin. One part lignolysate was diluted with five parts M9 minimal medium to generate the growth medium ($n = 4$). **h** Initial concentrations and consumption of major components of the lignolysate ($n = 4$). **i** Initial concentrations and consumption of minor/trace components of the lignolysate ($n = 4$). **j** Mevalonate titer over the course of bioproduction from formate-containing lignolysate, a maximum titer of 32 (±5.8) mg/L was achieved in small scale ($n = 4$). Data represent the mean ± s.d. of biological replicates. Source data are provided as a Source Data file.

A major advantage of the aerobic rGlyP is that microbial cell factories that employ this pathway have the ability to perform oxidative phosphorylation to generate ATP. Currently, state-of-the-art C1 bioproduction utilizes anaerobic acetogens such as *Clostridium autoethanogenum*[25,56,57]. While acetogens have been used industrially to produce organic acids, ketones, and alcohols at very high titers, rates, and yields, these organisms are extremely limited in their ability to synthesize products that require net ATP[25,26]. For example, anaerobic acetogenic bacteria have been unable to produce high titers of the

ATP-dependent class of natural products, terpenoids[26,27]. Terpenoid biosynthesis is achieved by the phosphorylation of mevalonate and subsequent decarboxylation and isomerization. Multiple mevalonate derived isoprene units can be combined to form mono-, sesqui-, di-, sester-, etc., terpenoids.

To demonstrate the utility of K4Me2 as a bioproduction platform for terpenoids, we engineered it to produce the hemiterpenoid isoprenol, which also has utility as a sustainable aviation fuel precursor[58]. We employed the highly efficient IPP bypass pathway, utilizing a

mutant phospho-mevalonate decarboxylase (PMD) that allows for the direct conversion of mevalonate-5-phosphate to isopentenyl monophosphate (IP)[59]. Isoprenol is then formed from the dephosphorylation of IP. To produce isoprenol, plasmid pBbA51a_isoprenol was constructed, and K4Me2 was transformed with this plasmid creating strain K4Me2_isop. K4Me2_isop was cultivated on e-formate in small-scale batch mode and reached a maximum isoprenol production titer of $11.9 \pm 5.1$ mg/L (Fig. 4c). Although a modest overall titer, the detection of isoprenol validates the feasibility of producing commercially valuable terpenoids from formate, and further strain and process optimization holds immense promise for improving production titers from formate in fed-batch mode. Additionally, the production of a fuel from e-formate represents the reversal of a combustion reaction, as $CO_2$ is converted into a flammable chemical (Fig. 4d). This highlights the potential for this technology to enable circular uses of carbon.

### Demonstration of bioproduction from formate and other acids derived from oxidative degradation of lignin

Despite its potential for sustainable, low-emission chemical production, bioproduction from e-formate or other electrochemical feedstocks is predicted to be relatively costly, limiting its near-term utility in the generation of low-cost, high-volume commodity biochemicals. This is mostly due to high inputs of electricity, which, even at low renewable electricity prices, will make it hard to compete with the prices of fossil-fuels and petrochemicals in the near future[9,45,46,60]. As an alternative to costly e-formate, organic waste sources could be considered as a sustainable and inexpensive source of formate. An abundant and inexpensive organic waste source that currently has limited routes to valorization is lignin. Lignin is a recalcitrant polymer that gives plant cell walls their mechanical strength and is the second most abundant biopolymer on Earth. The pulp and paper industries alone produce approximately 50 billion tons of lignin per year globally, 98% of which is either burned or disposed of in landfills[61]. A promising route to valorization is the oxidative degradation of lignin into formate or other bioavailable molecules[14].

Through oxidative degradation of lignin, we were able to produce a mixture of low molecular weight organic acids the most abundant of which was formate (Fig. 4h). Compared to other methods of lignin decomposition, this oxidative method requires substantially less energy input as it is catalyzed at much lower temperatures (140 °C compared to 250–800 °C)[62-64]. Additionally, this oxidative route produces soluble, bioavailable acids in contrast to other methods which produce hydrophobic lignin monomers or synthesis gas[62,65,66]. The other acids generated through our lignin degradation method can be catabolized by wild-type E. coli and other industrial hosts[67]. However, the formate (which is the most abundant product and constitutes 37% of the total bioavailable carbon identified in the mixture) cannot natively be used as a carbon source by common industrial hosts and therefore would be wasted or else accumulate to toxic concentrations in an industrial fermentation. Hence, we tested this lignolysate as a substrate for production in our formatotrophic, mevalonate-producing E. coli strain K4Me2_mev.

The lignolysate was diluted into M9 minimal medium and used as a carbon and energy source for the growth of K4Me2_mev (Fig. 4g) and the production of mevalonate. Over the course of the mevalonate bioproduction, all formate present in the culture was consumed (Fig. 4h) and a maximum mevalonate titer of $32.0 \pm 5.8$ mg/L was achieved over the course of a 240-hour cultivation (Fig. 4j). In addition to formate ($\approx 10$ mM), the lignolysate culture medium also contained acetate ($\approx 2$ mM) and glycolate ($\approx 1$ mM), as well as less than 1 mM each of gluconate, propionate, succinate, pyruvate, malate and tartrate, which were all fully consumed with the exception of tartrate (Fig. 4h, i). This small amount of gluconate likely arose from trace amounts of cellulose, which were subsequently oxidized during the depolymerization of lignin. Shikimate, alpha-ketoglutarate, and fumarate were

also identified at sub 20 µM levels and were consumed over the course of bioproduction (Fig. 4i). The titers and final optical density (OD) in this bioproduction experiment were comparable to those observed with 100 mM e-formate, despite a significantly lower initial formate concentration. This suggests that other acids contributed significantly to both growth and titer. The conversion of this lignolysate solution into mevalonate demonstrates a promising route to valorize waste lignin.

## Discussion

The rGlyP has been proposed to be a highly efficient, synthetic formate assimilation route[24]. However, despite the demonstration of full formatotrophic growth via the synthetic rGlyP in E. coli and C. necator and subsequent optimization and ALE, previous work has not managed to demonstrate doubling times faster than 6 h[18,19,33,34,37,68]. This is relevant as fast growth rates indicate rapid formate assimilation to chemical intermediates and substrates for bioproduction pathways; increasing the rate of product formation is an important metric for improving the economics of bioproduction.

Through the heterologous expression of a metal-dependent, multi-subunit formate dehydrogenase with a faster $k_{cat}$, we were able to achieve an increased growth rate compared to the original strain. A relatively short ALE campaign of that strain led to a greatly increased growth rate, very close to the fastest recorded formatotrophic growth in nature. The achieved doubling time of $\approx 4.5$ h falls in the same range as the recently established growth of synthetic methylotrophs growing on methanol, another promising C1 feedstock, putting synthetic cell factories for both these soluble C1 substrates on par[69-72]. Using this newly developed strain of E. coli capable of fast growth on formate, we were able produce an industrially relevant titer of mevalonate from formate alone, demonstrating that the production of multiple grams per liter of a bioproduct can be achieved from formate as sole carbon and energy feedstock.

Developing a strain of E. coli, an industrial host with widespread commercial application, into a robust formatotroph has direct biotechnological utility[73,74]. A formatotrophic E. coli could connect chemical processes that generate e-formate from $CO_2$ and electricity to established biological processes allowing for precision fermentation to generate chemicals, proteins, materials, and fuels ultimately from $CO_2$. This could allow for carbon-negative chemical biomanufacturing and/or circular uses of carbon. In this study, we found that a formatotrophic E. coli harboring cnFDH could allow for a more substantial redirection of biochemical flux into a bioproduct, mevalonate, at the expense of biomass accumulation. This was likely due to a greater capacity for strains harboring the cnFDH, which only occupies $\approx 0.2\%$ of the proteome (vs. $\approx 4\%$ for psFDH)[75], to reallocate proteome space for bioproduction routes.

This work can also impact biotechnological applications where supplementation of formate as an auxiliary energy substrate has the potential to improve the titer, rate, or yield of a given bioproduct. For example, formate has been used in mixotrophic bioproduction solely as an energy source, and this arrangement has allowed for increased product titers from other organic feedstocks[76-78]. These works have so far utilized slow, metal-independent FDH variants. The demonstrated functional expression of a fast, metal-dependent FHD could also improve the rates and efficiency of such cell factories in which formate is added as auxiliary energy substrate.

Furthermore, this work tested at small scale how the generated synthetic formatotrophic platform could perform for bioproduction with different renewable, formate-containing feedstock streams. We demonstrated the production of mevalonate and also isoprenol from e-formate. The production of the biofuel isoprenol is noteworthy as it represents the production of a terpenoid, a wide class of value added chemicals that cannot be easily produced from $CO_2$ using energy-efficient Wood-Ljungdahl anaerobes[26]. We introduce a method to

generate formate, alongside other feedstock organic acids, at potentially lower overall cost from abundant waste lignin. Cheaper formate feedstock could allow for economical production of lower value products such as commodity chemicals and fuels.

This work demonstrates fast and efficient biological conversion of formate to biomass and formate to product in a non-native formate consumer. Additionally, it explores an alternative $CO_2$-to-formate pathway using lignin that could enable more favorable economics for a formate bioeconomy. Together these results bring the promise of a sustainable formate bioeconomy closer to economic and technical feasibility.

## Methods

### DNA cloning and assembly

The genes *fdsGBACD* and their corresponding native RBSs were amplified by PCR from the genomic DNA of *Cupriavidus necator H16* using primers AC1 and AC2. Plasmid pBbS1k_RFP was amplified using primers AC3 and AC4. These DNA fragments were digested with BglII and MfeI and ligated with T4 DNA ligase to generate pBbS1k_cnFDH. Plasmid pBbA5a_mevalonate was constructed by the ligation of a XhoI and BglII digested DNA fragment, derived from plasmid JBEI-3093[76], containing the genes catalyzing the upper mevalonate pathway into a pBbA5a BglBrick backbone fragment. Plasmid pBbA51a_isoprenol was generated in two steps. The intermediate plasmid pBbA5a_7575 was constructed by ligation of a gene insert derived from EcoRI and BamHI digestion of JBEI_7575[79] into the pBbA5a backbone[50]. pBbA5a_7575 was then amplified with primers AC5 and AC6 to allow for the insertion of a mutant (R74G) phosphomevalonate dehydrogenase[59] and mevalonate kinase genes amplified from JBx_078554 with primers AC7 and AC8 using NEBuilder HiFi DNA Assembly Master Mix. Primer sequences can be found in Supplementary Table 4. All plasmids and their sequences are available for retrieval at Joint BioEnergy Institute's Inventory of Composable Elements (https://public-registry.jbei.org/login) according to the registry numbers found in Supplementary Table 5. Primers were synthesized by Integrated DNA Technologies.

### Genome engineering

Gene knockouts were performed by lambda red recombineering. Plasmid pSB54 containing a chloramphenicol resistance cassette was amplified by PCR using primers containing overlaps homologous to the chromosomal target. For deletion of the chromosomally integrated psFDH from K4e primers AC9 and AC10 were used. Allelic exchange was facilitated by the expression of the Red Disruption System from plasmid pKD46[80]. To remove the kanamycin cassette from Keio collection strain BW25115 *lpd::kan* plasmid pCP20 was used[80]. All strains used are shown in Supplementary Table 1.

### Growth medium and conditions

All strains were routinely cultured and propagated in LB medium (1% NaCl, 0.5% yeast extract, and 1% tryptone). BW25113 Δ*lpd* was cultivated in MOPS minimal medium (Teknova). The strain was first adapted in MOPS minimal medium containing 20 mM each glycerol, succinate, and acetate then washed three times (with MOPS medium without added carbon source) and inoculated into medium containing 20 mM acetate and 60 mM formate. K4e and K4M strains were cultivated in M9 medium (50 mM $Na_2HPO_4$, 20 mM $KH_2PO_4$, 1 mM NaCl, 20 mM $NH_4Cl$, 2 mM $MgSO_4$ and 100 μM $CaCl_2$) with trace elements (134 μM EDTA, 13 μM $FeCl_3 \cdot 6H_2O$, 6.2 μM $ZnCl_2$, 0.76 μM $CuCl_2 \cdot 2H_2O$, 0.42 μM $CoCl_2 \cdot 2H_2O$, 1.62 μM $H_3BO_3$, 0.081 μM $MnCl_2 \cdot 4H_2O$) with the addition of 25 μm $Na_2MoO_4$ (Millipore-Sigma)[18]. An intermediate minimal medium adaptation step was performed by a 1:100 inoculation of a LB-propagated K4 stain to M9 media containing 10 mM glucose, 30 mM formate, and 1 mM glycine. This culture was grown for 16 h and then washed three times (with M9 medium without added carbon source) before inoculation into fresh medium for

formatotrophic growth. Preculture and propagation medium contained kanamycin to maintain pBbS1k_cnFDH, however, kanamycin was not added to media containing only formate as the plasmid is maintained by its necessity for growth. For cultivation of K4 strains in a 10% $CO_2$ atmosphere during growth on formate, strains were grown in 18 × 150 mm borosilicate glass culture tubes (Chemglass). These tubes were sealed with blue chlorobutyl stoppers (Chemglass) and aluminum seals (Agilent) a volume of $CO_2$ corresponding to 10% of the headspace of the vessel was injected through the butyl stoppers with a syringe. For all experiments, analysis was conducted on independently growing cultures, and no data presented arise from the repeated assay of a single culture at a given time point. All chemicals used for growth and analysis were purchased from MilliporeSigma.

### Proteomic analysis

Protein was extracted from cell pellets, and tryptic peptides were prepared using a standard protocol[81]. Briefly, cell pellets were lysed in Qiagen P2 Lysis Buffer, followed by protein precipitation with NaCl and acetone. The protein pellet was washed, resuspended in ammonium bicarbonate/methanol, and quantified using the DC protein assay (BioRad, USA). Proteins were reduced with tris 2-(carboxyethyl)phosphine, alkylated with iodoacetamide, and digested overnight with trypsin at a 1:50 ratio.

Peptide samples were analyzed on an Agilent 1290 UHPLC system coupled to a Thermo Scientific Orbitrap Exploris 480 mass spectrometer using a C18 column and a 10 min gradient of solvent A (0.1% FA in H2O) and solvent B (0.1% FA in ACN). Data-independent acquisition (DIA) mode was used with 3 survey scans and 45 MS2 scans, covering $m/z$ 380–985[82].

DIA raw data were analyzed using DIA-NN in library-free mode with an *E. coli* Uniprot proteome database and heterologous protein sequences[83]. Automated mass tolerance and retention time optimization were employed, with protein quantification using the Top3 method[84,85]. DIA-NN results were filtered at a global FDR of 0.01 for precursor and protein group levels. Data have been deposited to ProteomeXchange (PXD059132)[86]. DIA-NN is available at Github (https://github.com/vdemichev/DiaNN).

### Determination of doubling time

For the determination of doubling time, K4 strains adapted to minimal M9 medium containing 30 mM formate, 10 mM glucose, and 1 mM glycine were washed three times and then inoculated into fresh medium containing 1 mM glucose and 60 mM formate in a sealed glass culture tube containing 10% $CO_2$ (air balance). This second adaptation was harvested during the mid-late exponential growth phase, washed three times, and used to inoculate a medium containing only formate at an initial $OD_{600}$ of 0.01. The growth of this culture was monitored by absorbance at 600 nm in a SPARK multimode microplate reader with $CO_2$ control at 10% (air balance). A total of 150 μL of culture was overlaid with 50 μL of mineral oil to prevent evaporation while allowing gas exchange. OD values measured in the plate reader were calibrated to represent OD values in standard cuvettes, according to the following equation:

$$OD_{cuvette} = OD_{plate}/0.23 \qquad (1)$$

All growth experiments were repeated independently at least 3 times to ensure reproducibility. A detailed protocol for growth measurements with a microplate reader is provided in Wenk et al.[87] Python scripts for doubling time calculations are available at Github [https://github.com/he-hai/growth2fig], originally published by He et al.[88].

### Determination of biomass yield

K4e and K4Me2 strains in log phase growth on 90 mM formate were inoculated (1:1000) into a fresh 50 mL of M9 mineral medium

containing 90 mM formate and incubated at 37 °C with shaking at 120 rpm and 10% $CO_2$ atmosphere until late exponential growth was reached. The resulting cellular biomass was pelleted and washed twice with sterile distilled water. The supernatant containing un-metabolized formate was saved for HPLC analysis. Pelleted cells were resuspended in 400 μL of distilled water and left to dry for 24 h at 100 °C in pre-weighed aluminum weighing boats. Dried cellular biomass was weighed, and formate consumed in each sample was analyzed by HPLC (see methods for analysis of organic acid feedstock). Biomass yield is calculated as the ratio of cellular biomass to moles of formate consumed.

### Adaptive laboratory evolution and whole genome sequencing

Adaptive laboratory evolution was conducted by passaging the strain K4M in a M9 mineral medium containing 60 mM formate. Passaging was done by a 1:10 dilution of the K4M strain whenever the optical density of the culture exceeded 0.4 and continued until the lag phase was noticeably reduced which required approximately 25 doublings. Single clones of the evolving culture were isolated at this time point to identify the mutations responsible for the improvement of growth characteristics. Plasmids contained in these strains were sequenced using Oxford Nanopore technology (Plasmidsaurus). The whole genome of these strains was also sequenced using a method adapted from Zhan et al.[89] Genomic DNA was isolated using a Wizard genomic DNA purification kit. 100 ng of isolated DNA was fragmented using the xGen™ DNA Library Prep EZ kit (Integrated DNA Technologies, Inc.), and 600 bp fragments were enriched using SPRI beads (Beckman Coulter). The fragments were treated with end-repair, A-tailing, and ligation of Illumina-compatible adapters (Integrated DNA Technologies, Inc.) using the xGen™ DNA Library Prep EZ kit. Bioanalyzer High Sensitivity DNA Kit (Agilent) and Qubit Fluorometers (ThermoFisher Scientific) were used to determine the concentration of the libraries. Libraries were sequenced on the Illumina Miseq.

Whole genome sequencing reads were filtered and trimmed using AfterQC version 0.9.7[90]. Reads were then mapped to the *E. coli* BW25113 reference genome (CP009273.1) using the breseq pipeline version 0.33.1[91]. Consensus frequency cutoff was set to 0.75, and required match fraction set to 0.95. The breseq pipeline was run in clonal or population mode depending on the sample type. Mutation analysis was performed using the workflow implemented on ALEdb[92]. On average, a coverage depth of 169 reads per bp was utilized to make mutation calls for sequenced strains.

### Bioproduction

For bioproduction of mevalonate at small scale, fresh transformants were prepared by transformation of pBbA5a_3093 into K4e and/or K4M strains as indicated. Transformants were cultivated in LB overnight at 37 °C then diluted 1:100 into M9 mineral medium containing 30 mM formate, 10 mM glucose and 1 mM glycine and incubated for 16 h at 37 °C. This preculture was then washed 3 times with M9 medium without any carbon source and used to inoculate another preculture media containing the same carbon sources as the final production media, but without any inducer at an initial $OD_{600}$ of 0.1. Once this culture was actively growing in mid-log phase it was harvested, washed three times, and used to inoculate the final bioproducion experiment containing the indicated carbon sources, at the indicated initial $OD_{600}$ with expression of mevalonate genes induced by the introduction of 0.5 mM IPTG.

For the production of isoprenol the same procedure was followed with plasmid pBbA51a_isoprenol. However, IPTG was initially omitted from the culture to allow for cell growth in the absence of bioproduction burden. IPTG was then added at 72 h with additional formate (as indicated).

For fed-batch bioproduction, precultures were inoculated into 150 mL M9 medium containing 30 mM sodium formate in 250 mL

GLS80 flasks (Duran) at 37 °C and at a final $OD_{600}$ of 0.5 under a 10% $CO_2$ atmosphere. The cultures were stirred at 1000 rpm with a magnetic stirrer (10 mm diameter, 60 mm length), and the pH was controlled to 7.0 by the addition of 96% formic acid using the Bluelab pH Controller Connect (Bluelab). After 16 h of culture, the ammonium concentration was raised to 30 mM. Formate and ammonium concentrations were adjusted to 30 mM twice a day. When the $OD_{600}$ reached a value of 10, mevalonate production was induced by the addition of 0.5 mM IPTG.

### Quantification of bioproducts

For quantification of mevalonate in the g/L range from fed-batch bioconversion HPLC-RID was employed due to its wide linear range in those concentrations. An Agilent series 1200 was used to deliver a 50 μL injection volume of clarified culture supernatant to an Aminex HPX-87H column (BioRad) at 65 °C with subsequent detection of mevalonate by refractive index detector (RID) maintained at 50 °C. This separation and quantification were performed in a 5 mM $H_2SO_4$ solution flowing at a rate of 0.600 mL/min.

For quantification of mevalonate in the low-middle mg/L range (i.e., from small-scale experiments) LCMS was employed due to its low limit of detection and high accuracy at low concentrations. An Agilent 1260 Infinity II LC system equipped with a MSD/iQ mass spectrometer was used to deliver a 10 μL injection volume of clarified culture supernatant to an Astec® C18 HPLC Column (5 μm particle size, L × I.D. 15 cm × 4.6 mm) (Supelco) for separation at room temperature with subsequent detection by MSD/iQ mass spectrometer. Ultrapure water and methanol (MeOH) containing 0.1% formic acid were used as the mobile phase in the following gradient 5% MeOH to 85% MeOH over 4 min, 85% MeOH to 98% MeOH over 3.8 min both at a flow rate of 0.420 mL/min then 98% MeOH to 5% MeOH over 0.4 min and hold at 5% MeOH for 4.8 min at a flow rate of 0.650 mL/min. Mevalonate was monitored by a negative mode single-ion monitoring (SIM) at $m/z = -147$.

Isoprenol was extracted from bacterial culture by mixing with an equal volume of isoprenol containing 30 mg/L of butanol internal standard. Extraction was performed by vigorous vortexing for 15 minutes followed by centrifugation at $13,000 \times g$ for 5 min. The organic layer was then isolated and analyzed using an Agilent Technologies Intuvo 9000 GC System and 5977B MSD equipped with a 15 m × 0.25 mm, 0.25 μm Agilent DB-WAX UI column. A 1 μL volume was injected. The temperature program of the oven was as follows: initial temperature 50 °C, ramp from 50 to 150 °C at 10 °C/min, ramp from 150 °C to 230 °C at 100 °C/min, and then hold at 230 °C. Signal intensity was compared to an authentic standard to determine production titer and was normalized to the signal intensity of the 1-butanol internal standard. Mass spectra were compared to an authenticated standard.

For analysis of the mass spectra of mevalonate, bioproduced mevalonate was derivatized to mevalonolactone to decrease the polarity of the molecule and make it compatible with GC-MS analysis. A derivatization protocol was adapted from Satowa et al.[93] 50 μL of HCL was added to 300 μL culture media. Samples were shaken vigorously at 60 °C for 30 min. Mevalonolactone was then extracted with 700 μL ethyl acetate and analyzed by GC-MS using the same procedure as for isoprenol. Mass spectra were compared to an authenticated standard (Supplementary Fig. 5).

### Statistical methods

All statistical analysis reported in this work were performed by a two tailed *t*-test. All significant differences have *p*-values below 0.05. Error is presented as one standard deviation. Linear regression was performed to determine the correlation between final titer and growth rate or AtoB expression. Where applicable, sample size (n) and standard error of the mean (SEM) were incorporated into the calculation of

the coefficient of determination ($R^2$) to account for measurement variability. Error bars represent one standard deviation.

## Electroreduction of $CO_2$ to formate

A dual-chamber microbial electrosynthesis cell was constructed using Plexiglass tubes. The system employed a three-electrode configuration, consisting of a titanium electrode coated with a 10 μm layer of ruthenium-iridium (American Elements, USA) as the anode and a carbon fiber paper electrode coated with a bismuth catalyst (Dioxide Materials, USA) as the cathode. Both anode and cathode had an identical geometric area of 2 cm². The anode and cathode chambers were separated by an alkaline anion exchange membrane (Sustainion X37-50 Grade T, Dioxide Materials, USA) with a total surface area of 12.5 cm², arranged in a sandwich-type configuration.

The cathode chamber contained a Ag/AgCl reference electrode (3 M KCl, MF-2056, BASi, USA) positioned within 1 cm of the cathode electrode. A potential of −2 V vs. Ag/AgCl was applied to the cathode using a multi-channel potentiostat (VMP-3e, BioLogic, USA). Both the anode and cathode chambers were filled with 50 mM $KHCO_3$ as the electrolyte, with identical working volumes of 35 mL. $CO_2$ was continuously sparged into the cathode chamber at a flow rate of 15 mL/min (20 °C, 1 bar) through a diffuser facing the cathode, with flow rate control by a gas flow meter (Dakota Instruments, USA). The anolyte was recirculated using a peristaltic pump from a 300 mL reservoir at a flow rate of 0.5 mL min$^{-1}$ and sparged continuously with high purity nitrogen (99.99%) to flush $O_2$ out of the anode chamber. The experiment was maintained at 30 °C in an incubator, with continuous mixing provided by a magnetic stir bar at a fixed rotation rate. After five days of operation, liquid products were collected from the catholyte for formate analysis.

## Generation of lignin-rich solids

Cholinium lysinate ([Ch][Lys])-based pretreatment of poplar biomass and subsequent saccharification was performed in a one-pot configuration in a 1 L Parr 4520 series bench top reactor (Parr Instrument Company, Moline, IL) as described elsewhere[94] with slight modification. Briefly, poplar, [Ch][Lys], and water were mixed in a 3:2:15 ratio (w/w) in the Parr vessel to achieve 15 wt. % solids loading. This slurry mixture was pretreated for 3 h at 160 °C with stirring at 80 rpm powered by process (Parr Instrument Company, model: 4871, Moline, IL) and power (Parr Instrument Company, model: 4875, Moline, IL) controllers using three-arm, self-centering anchor with PTFE wiper blades. After 3 h, the pretreated slurry was cooled down to room temperature. The pH of the cold, pretreated mixture was adjusted to 5 with concentrated hydrochloric acid. Enzymatic saccharification was carried out at 50 °C for 72 h at 80 rpm using enzyme mixtures Cellic CTec3 and HTec3 (9:1 v/v) at a loading of 20 mg protein per g biomass. After 72 h, polysaccharides in the poplar biomass were hydrolyzed into monomeric sugars. The slurry was centrifuged for solid-liquid separation followed by water washing until the pH of the colorless washing was neutral. The washed material was freeze dried to obtain lignin-rich solids.

## Catalytic oxidation of lignin-rich solids into aliphatic acids

Lignin-rich solids, polyoxotungstate catalyst, hydrogen peroxide, and water were mixed, pressurized with 100 psi $N_2$, and heated at 140 °C for 1 h. After 1 h, unreacted solids were separated from the aqueous stream containing depolymerized bioavailable products by centrifugation[14]. The aqueous stream containing depolymerized products was diluted and filtered through 0.45 μm SFCA sterile filter units. The filtered lignolysate was used for bioproduction studies.

## Analysis of organic acid feedstock

Organic acids in lignolysate (with the exception of acetate and formate which are more appropriately analyzed by HPLC-RID) were measured using reversed-phase chromatography and high-resolution mass spectrometry[95]. Liquid chromatography (LC) was performed on an Ascentis Express RP-Amide column (150 mm × 4.6 mm, 2.7 μm particle size) with an Agilent 1290 Infinity II UHPLC system. A 2-μL injection volume was used, with the sample tray and column compartment set to 4 °C and 50 °C, respectively. Mobile phases consisted of 0.1% formic acid/84.9% water/15% methanol (A) and 0.04% formic acid/5 mM ammonium acetate in methanol (B). Metabolites were separated via gradient elution: 0–30% B in 5.5 min, 30–100% B in 0.2 min, held at 100% B for 2 min, returned to 0% B in 0.2 min, and equilibrated at 0% B for 2.5 min. The flow rate was 0.4 mL/min, increasing to 0.65 mL/min at 7.7 min and held for 2.5 min, with a total LC runtime of 10.4 min. The HPLC was coupled to an Agilent 6545 QTOF-MS.

For quantification of formate and acetate, which could not be detected by LC-MS, HPLC-RID was employed. An Agilent series 1200 was used to deliver a 50 μL injection volume of clarified culture supernatant to an Aminex HPX-87H column (BioRad) at 65 °C with subsequent detection of formate and acetate by refractive index detector (RID) maintained at 50 °C. This separation and quantification were performed in a 5 mM $H_2SO_4$ solution flowing at a rate of 0.600 mL/min.

## Reporting summary

Further information on research design is available in the Nature Portfolio Reporting Summary linked to this article.

## Data availability

Raw sequencing data have been deposited to the Genome Sequence Archive (GSA) with accession code CRA023445. Raw proteomic data have been deposited to the PRIDE database with accession code PXD062361. Source data are provided with this paper.

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

## Acknowledgements

We thank Rita Kuo and Joshua McCauley for the development and operation of the whole-genome sequencing pipeline. We thank Muyao Wu for the development and operation of the ALEdb database. This work was supported by the US Department of Energy (DOE) Joint BioEnergy Institute (https://www.jbei.org), supported by DOE, Office of Science, Biological and Environmental Research Program, under contract DEAC02-05CH11231 between DOE and Lawrence Berkeley National Laboratory (J.D.K., C.J.P., B.A.S., A.M.F.), the DOE Distinguished Scientist Fellow Program (J.D.K.), the Philomathia Foundation (J.D.K.), National Institutes of Health grant R01 AT010593-02 (J.D.K.), and National Science Foundation grant 2036849 (J.D.K.). V.R. and N.J.C. acknowledge funding by VLAG Open Round by the VLAG Graduate School. Sandia National Laboratories is a multi-mission laboratory managed and operated by National Technology and Engineering Solutions of Sandia, LLC, a wholly owned subsidiary of Honeywell International Inc., for the U.S. Department of Energy's National Nuclear Security Administration under contract DE-NA0003525. This paper describes objective technical results and analysis. Any subjective views or opinions that might be expressed in the paper do not necessarily represent the views of the U.S. Department of Energy or the United States Government.

## Author contributions

Conceptualization: A.E.C.; Methodology: A.E.C.; Investigation: A.E.C., M.H., V.R., F.C., H.C., B.S.Z, G.G.B., D.N.C., M.G., J.W.G., B.C., Y.C., E.A.T, E.E.K.B., C.J.P., A.M.F., S.T., F.K. and B.A.S.; Writing Original Draft: A.E.C.; Writing Review and Editing: A.E.C., J.D.K., D.N.C., A.M.F., V.R. and N.J.C.; Resources and Supervision: J.D.K., N.J.C. All authors revised and approved the manuscript.

## Competing interests

J.D.K. has financial interests in Ansa Biotechnologies, Apertor Pharma, Berkeley Yeast, BioMia, Cyklos Materials, Demetrix, Lygos, Napigen, ResVita Bio, and Zero Acre Farms. F.C. and F.K. are employed by b.fab GmbH, a German biotech company aiming for the biomanufacturing of proteins and chemicals from C1 feedstocks. B.A.S. has financial interests in Illium Technologies, Caribou Biofuels, and Erg Bio. The other authors declare no competing interests.

## Additional information

¹Joint BioEnergy Institute, Emeryville, CA, USA. ²Department of Molecular and Cell Biology, University of California, Berkeley, Berkeley, CA, USA. ³Laboratory of Microbiology, Wageningen University and Research, Wageningen, The Netherlands. ⁴b.fab GmbH, Cologne, Germany. ⁵Department of Bioresource and Environmental Security, Sandia National Laboratories, Livermore, CA, USA. ⁶Advanced Biofuels and Bioproducts Process Development Unit, Emeryville, CA, USA. ⁷Biological Systems and Engineering, Lawrence Berkeley National Laboratory, Berkeley, CA, USA. ⁸Department of Bioengineering, University of California, San Diego, CA, USA. ⁹The Novo Nordisk Foundation Center for Biosustainability, Technical University Denmark, Kemitorvet, Kongens Lyngby, Denmark. ¹⁰QB3 Institute, University of California, Berkeley, CA, USA. ¹¹Department of Chemical and Biomolecular Engineering and Department of Bioengineering, University of California, Berkeley, CA, USA. ✉e-mail: aidancowan@berkeley.edu; keasling@berkeley.edu; nico.claassens@wur.nl

