## [Peer Review file · Nature Communications]

Fast growth and high-titer bioproduction from renewable formate via metal-dependent formate dehydrogenase in *Escherichia coli*

Corresponding Author: Dr Nico Claassens

Version 0:

Reviewer comments:

Reviewer #1

(Remarks to the Author)

The authors report the achievement of an important milestone on the path to electricity based microbial production. It is an important step forward which some groups tried to take in the past but failed and it will be useful to the entire community. The paper is generally clear, convincing and to the point

The few comments remaining are below:

The authors claim that K4e did not achieve the high titers of Mevalonate as K4Me2 and that proves that psFDH is inferior to cnFDH. However, as the authors state K4e was able to evolve further to achieve a doubling time of 6h, so there is still some potential to improve the yield which was not realized in this study. If the max growth rate of each directly correlates with the max titer, that would be somewhat more convincing since K4Me2 has the highest growth rate. Another option would be to test the Mevalonate yield of the evolved K4e strain with the higher growth rate.

In the last results section, the authors report the production of mevalonate from oxidative degradation of lignin using the engineered strain. However, it is unclear what the advantage of this strain is over wild-type *E. coli* expressing the pBbA5a_mevalonate plasmid. Could you clarify how much mevalonate was produced from formate versus the other acids?

In the introduction, when referring to the WL pathway, the authors wrote that it has a low ATP cost but: "this efficiency is from necessity as this anaerobic pathway is extremely limited by its energetics and its ability to produce ATP ... cannot be used to efficiently produce..."

This could be misleading. The WL pathway is definitely the most efficient in fixing CO₂ (in terms of ATP). However, since it is anaerobic and does not exist in phototrophic organisms, it must also provide the ATP for production (by fermenting some of the acetyl-CoA produced by it). For a fair comparison to the Calvin and serine cycles, one should assume ATP is provided from another source. Perhaps it would be clearer to say that WL is less efficient because it is oxygen-sensitive.

P.3, "there are three dominant formatrophic pathways: the Calvin cycle..."  is the Calvin cycle a formatrophic pathway?

P. 3 "doubling time of ~4.5 hours" the more mathematically accurate symbol is "doubling time of ≈4.5 hours"

The last sentence in page 8: "This is a highly desirable trait for strains used in industrial applications that seek to maximize titer and yield." refers to the lower yield and higher titer. However, this is an almost universal feature of any cell due to the stoichiometric tradeoff between the biomass rate and the secretion of byproducts (for the same uptake rate). The phrasing could be improved to explain better what is special in this case.

In Figure 3a the stoichiometry seems wrong for FDH, it produces one NADH per formate and not two.

In the section about bioproduction from lignin via formate, the authors conclude by "The conversion of this lignin solution into mevalonate demonstrates a promising route to valorize waste lignin.". A bit more context would be helpful, as

this is not compared to any other known methods for valorizing lignin (e.g. syngas).

Reviewer #2

(Remarks to the Author)

This study describes the construction of an Escherichia coli strain capable of rapid growth using formate as both a carbon and energy source. A key issue in this study is that the low activity of formate dehydrogenase (FDH) results in limited growth and energy supply. To address this, the researchers engineered an NADH-dependent E. coli strain and introduced a metal-dependent FDH, followed by adaptive laboratory evolution (ALE), achieving a significant reduction in doubling time. Using the obtained mutant plasmids, they demonstrated the production of mevalonate and isoprenol.

1. Through a short period of ALE, a mutant of the lac operator was obtained. What factors contributed to the rapid acquisition of this mutation? Please provide an explanation.

2. Introducing this mutant plasmid into the K4e psfdh::cat strain resulted in a shortened doubling time. The authors claim that this is due to an increased expression level of cnFDH, as shown by proteomic analysis. This suggests that similar results could be achieved by changing the appropriate promoter. It is recommended to verify this by conducting experiments using different promoters.

3. There is no discussion regarding Figure 2b, making it unclear what the authors aim to demonstrate with this figure.

4. Similarly, Figure 2e appears to lack discussion in the main text.

5. Figure 3c: The proteomic analysis indeed indicates a flux shift toward the mevalonate production pathway. The manuscript states,

“The use of cnFDH as an energy generation system may have allowed for reallocation of the ~4% of the proteome (Supplementary Fig. 2c) which was previously dedicated to psFDH to mevalonate pathway expression, allowing for improved conversion of formate to mevalonate in strains with the cnFDH energy module.”

However, the driving force directing the flux toward mevalonate production remains unclear. If this is determined by the expression level of cnFDH, similar results may be obtained by investigating the promoter or expression levels of cnFDH.

6. If e-formate and formate are chemically the same compound, then obtaining the same results is self-evident. If e-formate has any potential inhibitory effects on growth or production, please provide a description of this.

7. The motivation behind using lignin as a substrate is unclear. While it does contain other organic acids, this study primarily focuses on formate, making the inclusion of lignin seem somewhat misaligned with the study's main focus.

8. This study appears to address the issue of low FDH activity by increasing FDH expression levels. However, this does not necessarily solve the fundamental issue of low specific activity. Additionally, despite the increased expression levels, it is unclear why the flux is directed more toward product formation rather than growth. A discussion on this point would be beneficial.

Version 1:

Reviewer comments:

Reviewer #1

(Remarks to the Author)

The revised version of the manuscript looks great and there are no further comments.

Reviewer #2

(Remarks to the Author)

Thank you for your detailed response to my comments. The authors have clearly answered my questions.

REVIEWER COMMENTS

Reviewer #1 (Remarks to the Author):

The authors report the achievement of an important milestone on the path to electricity based microbial production. It is an important step forward which some groups tried to take in the past but failed and it will be useful to the entire community.

The paper is generally clear, convincing and to the point

Thank you for the time and effort you have taken to carefully review our manuscript. We are pleased to know you agree that this work is an important step forward and useful for a wide community.

We have carefully taken your comments and suggestions into account as we revised our manuscript. Your feedback has helped us improve the clarity and quality of this paper.

The few comments remaining are below:

The authors claim that K4e did not achieve the high titers of Mevalonate as K4Me2 and that proves that psFDH is inferior to cnFDH. However, as the authors state K4e was able to evolve further to achieve a doubling time of 6h, so there is still some potential to improve the yield which was not realized in this study. If the max growth rate of each directly correlates with the max titer, that would be somewhat more convincing since K4Me2 has the highest growth rate. Another option would be to test the Mevalonate yield of the evolved K4e strain with the higher growth rate.

This is an insightful comment, and we have now updated the text (Methods and Results) as well as Supplementary Figure 3 to include the suggested correlation analysis. In Supplementary Figure 3 we have now included the results of a regression analysis between mevalonate titer and growth rate (from Fig 2d), as well as a regression analysis between mevalonate titer and proteome fraction devoted to AtoB, which catalyzes the first dedicated step of the Mev pathway (from Figure 3a). They both show a strong correlation.

To introduce these analyses we have updated the manuscript to include this text:

“Interestingly, the final titers achieved for each strain were highly correlated with the expression level of atoB (Supplementary Fig. 3c), the first dedicated step in the mevalonate pathway. It stands to reason that the expression level of atoB may dictate the metabolic flux which is diverted from biomass accumulation to mevalonate and that increased expression could increase flux into the mevalonate pathway. Final titer was also correlated with doubling time, though doubling time, which is ultimately dependent on the rate of formate assimilation, would be most expected to correlate with productivity (which was also increased in K4Me2 relative to K4e (Supplementary Fig. 4c)).”

In the last results section, the authors report the production of mevalonate from oxidative

degradation of lignin using the engineered strain. However, it is unclear what the advantage of this strain is over wild-type *E. coli* expressing the pBbA5a_mevalonate plasmid. Could you clarify how much mevalonate was produced from formate versus the other acids?

The advantage of using K4Me2 for bioproduction is that it is able to utilize formic acid, which is present at a high molar amount relative to the other organic acids present in the solution. The other acids can indeed be consumed by wild-type *E. coli*, however without consuming formate the strain would miss out on $\approx 37\%$ of the bioavailable carbon identified in the lignolysate mixture and in an industrial setting the accumulation of unmetabolized formate could hinder growth.

To clarify the advantage of using an organism capable of using formate as a carbon and energy source the following text has been added to the manuscript. Additionally the contribution of formate to the overall pool of bioavailable carbon has been expressly stated to clarify its contribution:

“The other acids can be catabolized by wild-type *E. coli* and other industrial hosts.⁶⁶ However, the formate (which is the most abundant product and constitutes 37% of the total bioavailable carbon identified in the mixture) can not natively be used as carbon source by common industrial hosts and therefore would be wasted or else accumulate to toxic concentrations in an industrial fermentation.”

Furthermore the paragraph has been edited to better explain the main reason for including this section in the manuscript. This reasoning is based on the fact that electrochemically derived formate is relatively costly and that waste biomass derived formate may allow for a more realistic economic model for formatotrophic bioproduction of commodity chemicals in the near term:

“Despite its potential for sustainable, low-emission chemical production, bioproduction from e-formate or other electrochemical feedstocks is predicted to be relatively costly, limiting its near-term utility in the generation of low-cost, high-volume commodity biochemicals.”

“As an alternative to costly e-formate, organic waste sources could be considered as a sustainable and inexpensive source of formate.”

In the introduction, when referring to the WL pathway, the authors wrote that it has a low ATP cost but: "this efficiency is from necessity as this anaerobic pathway is extremely limited by its energetics and its ability to produce ATP ... cannot be used to efficiently produce..."

This could be misleading. The WL pathway is definitely the most efficient in fixing CO₂ (in terms of ATP). However, since it is anaerobic and does not exist in phototrophic organisms, it must also provide the ATP for production (by fermenting some of the acetyl-CoA produced by it). For a fair comparison to the Calvin and serine cycles, one should assume ATP is provided from another source. Perhaps it would be clearer to say that WL is less efficient because it is oxygen-sensitive.

The phrasing of these sentences has been updated to more accurately characterize the energetic constraints and oxygen-sensitivity of organisms utilizing the WL pathway. The text was updated to convey that while the WL pathway is an effective way of anaerobically producing some non-ATP-dependent products, terpenoids (like isoprenol) may be more suited for production in an aerobic host and with an oxygen-tolerant pathway like the one we describe in this work.

P.3, “there are three dominant formatotrophic pathways: the Calvin cycle...”  is the Calvin cycle a formatotrophic pathway?

The Calvin Cycle can be used for growth on formate as a carbon and energy source by oxidation of formate to generate NADH and CO₂. Using the energy from NADH, formate-derived CO₂ can be assimilated into metabolism. So, even though the Calvin cycle is not strictly a direct pathway for formate assimilation, but for CO₂, it is in nature used by several organisms (including *C. necator*) as the core assimilation pathway during formatotrophic growth. Hence, the text has been updated with alternative terminology: ‘pathways supporting growth on formate’ instead of ‘assimilating formate’.

“In nature, there are three dominant metabolic pathways supporting growth on formate: the Calvin cycle, the serine cycle, and the Wood-Ljungdahl pathway.²²”

P. 3 “doubling time of ~4.5 hours” the more mathematically accurate symbol is “doubling time of ≈4.5 hours”

Thank you for your attention to this detail. The manuscript has been updated and other instances of the use of a tilde in this context have been replaced with “≈”.

The last sentence in page 8: "This is a highly desirable trait for strains used in industrial applications that seek to maximize titer and yield." refers to the lower yield and higher titer. However, this is an almost universal feature of any cell due to the stoichiometric tradeoff between the biomass rate and the secretion of byproducts (for the same uptake rate). The phrasing could be improved to explain better what is special in this case.

This sentence has been reworded to acknowledge that this is a widely recognized, fundamental stoichiometric trade-off, which is desirable for biotechnological applications.

We reworded as follows:

“A trade-off between growth and product titer is a well-recognized characteristic of many bioproduction chassis. However, optimizing for production at the expense of growth can be difficult to engineer in microbial systems which are under constant selective pressure for improved growth.

In Figure 3a the stoichiometry seems wrong for FDH, it produces one NADH per formate and not two.

Thank you for your careful examination of this figure. The stoichiometry has been updated.

In the section about bioproduction from lignin via formate, the authors conclude by "The conversion of this lignolysate solution into mevalonate demonstrates a promising route to valorize waste lignin.". A bit more context would be helpful, as this is not compared to any other known methods for valorizing lignin (e.g. syngas).

More context was added to this section to briefly explain the benefits of our proposed technology compared to other methods.

We added the following sentences: "Compared to other methods of lignin decomposition this oxidative method requires substantially less energy input as it is catalyzed at much lower temperatures (140 °C compared to 250-800 °C).⁶¹⁻⁶³ Additionally this oxidative route produces soluble, bioavailable acids in contrast to other methods which produce hydrophobic lignin monomers or synthesis gas.^{61,64,65}"

Reviewer #2 (Remarks to the Author):

This study describes the construction of an Escherichia coli strain capable of rapid growth using formate as both a carbon and energy source. A key issue in this study is that the low activity of formate dehydrogenase (FDH) results in limited growth and energy supply. To address this, the researchers engineered an NADH-dependent E. coli strain and introduced a metal-dependent FDH, followed by adaptive laboratory evolution (ALE), achieving a significant reduction in doubling time. Using the obtained mutant plasmids, they demonstrated the production of mevalonate and isoprenol.

Thank you for your careful review of our manuscript. Your feedback has been invaluable to help us improve the clarity and narrative of this work. Each comment has been carefully considered and our responses can be found below.

1. Through a short period of ALE, a mutant of the lac operator was obtained. What factors contributed to the rapid acquisition of this mutation? Please provide an explanation.

The text has been updated to include an explanation for the rapid acquisition of this mutation to the lac operator which increased expression of cnFDH. Briefly, we believe that given the very slow growth rate of the initial K4M strain on formate, mutations which allowed for tuning FDH expression to the appropriate level had a large impact on growth rate and could rapidly outcompete the WT strain during the ALE.

We included the following explanation:

“The initial growth of K4M before FDH expression tuning was likely severely hindered by low levels of this key enzyme, which is required for energy generation (Supplementary Fig. 2a). Hence, single mutations that allow for the appropriate level of expression of the FDH enzyme lead to a large increase in growth rate. This explains why this mutation tuning the expression of the cnFDH promoter emerged rapidly over a short ALE campaign. ”

2. Introducing this mutant plasmid into the K4e psfdh::cat strain resulted in a shortened doubling time. The authors claim that this is due to an increased expression level of cnFDH, as shown by proteomic analysis. This suggests that similar results could be achieved by changing the appropriate promoter. It is recommended to verify this by conducting experiments using different promoters.

We originally chose the TRC promoter because it was the strongest (non-T7) promoter available to us through the BGL brick collection. Despite its strength, the strain needed some fine tuning of expression using ALE to find a more optimal expression level. However, the final optimal expression of cnFDH from this promoter is still much lower than the original psFDH, in accordance with the much faster kinetics of cnFDH necessitating less of the proteome. We have modified the text to highlight the importance of using ALE to tune the expression of cnFDH after the replacement of psFDH, rather than highlighting general increases in expression. ALE allowed the cell to approach an optimal level of cnFDH expression, which is likely difficult to achieve with rational design, especially considering we started with the strongest promoter in our commonly used BGL brick collection. We also added reference to the original Kim, 2020 publication which employed a similar strategy of estimating the appropriate promoter and then using a short ALE to tune expression to the appropriate level which was then stable over a much longer ALE (Kim, 2023). We can not exclude the possibility that rational design may have been able to immediately realize the appropriate “tuned” expression level, however we were not able to achieve fast growth by using other promoters in the BGL brick collection, nor by increasing the copy number of the TRC vector, nor by using the native cnFDH promoter during the initial phases of this study. We hope that the updated text will now highlight the importance of using ALE for facile expression tuning as demonstrated in previous publications and also make clear that the “tuned” expression level of cnFDH is far below the “tuned” expression level of psFDH, which it is replacing.

We added the following text to better explain our reasoning behind the design choices we made:

“We expect that ALE is the best method to enable rapid fine-tuning of cnFDH levels to optimal levels for fast growth. Tuning of FDH expression through short term ALE was also necessary to achieve faster growth of the original K4e strain and is therefore important to compare these two strains.¹⁸ We selected for fast formatrophic growth by ALE in the absence of IPTG, to enable formatotrophic growth without the need to add an inducer to the culture medium.”

Furthermore, the text has been improved to better convey the key finding which is that once expression has been optimized by ALE (in K4M* containing the reverse engineered expression-tuned cnFDH expression cassette) growth on formate is improved and cnFDH

expression is greatly decreased relative to psFDH expression in K4e. Wherever possible, the phrase “increasing expression” has been replaced with “expression tuning” to highlight the fact that the cell must have an appropriate expression of the given FDH.

3. There is no discussion regarding Figure 2b, making it unclear what the authors aim to demonstrate with this figure.

Thank you for pointing this out that the clarity of this section could be improved by adding more direct references to figure panels. Reference to figure 2b, visually summarizing the mutations observed in the evolved strains was added to aid in clarity. Additionally reference was made to figure 2c, showing the growth curves used to extract doubling times calculated in 2d. Reference in the text to 2d was also added where appropriate.

The following text was added:

“The mutations associated with a given strain are shown visually in Figure 2b and in tabular form in Supplementary Table 1.”

“The three K4M-based strains all grew faster than the original K4e strain and reached their final OD in less time (Fig. 2c, Fig. 2d).”

4. Similarly, Figure 2e appears to lack discussion in the main text.

To improve the clarity of the expression tuning narrative Fig. 2e is now referenced in the penultimate sentence of the formatotrophic growth section. We also added reference to Supplementary Fig. 2c which shows the expression of cnFDH in the evolved K4M clones compared to psFDH expression in the original K4e strain. We believe these modifications may help improve the clarity of this section and better direct the reader towards the data that support our claims. Thank you for your suggestions.

The following text was added:

“Despite a key mutation (lacO nt. 8 G to A) in the evolved strains which increased the expression of cnFDH from its untuned level in K4M (Fig. 2e), the overall expression of the cnFDH complex is still twenty-fold lower than the expression of psFDH in the K4e strain (Supplementary Fig. 2c), imparting less expression burden and potentially explaining the decreased doubling time.”

5. Figure 3c: The proteomic analysis indeed indicates a flux shift toward the mevalonate production pathway. The manuscript states,

“The use of cnFDH as an energy generation system may have allowed for reallocation of the ~4% of the proteome (Supplementary Fig. 2c) which was previously dedicated to psFDH to mevalonate pathway expression, allowing for improved conversion of formate to mevalonate in strains with the cnFDH energy module.”

However, the driving force directing the flux toward mevalonate production remains unclear. If this is determined by the expression level of cnFDH, similar results may be obtained by investigating the promoter or expression levels of cnFDH.

To clarify and support our hypothesis that the expression level of the mevalonate pathway dictates the relative fraction of formate that is converted to mevalonate at the expense of biomass formation we employed correlation analysis. More specifically we hypothesize that the expression level of *atoB* (the enzyme catalyzing the first step in the mevalonate pathway) dictates the fraction of formate which is diverted from central metabolism and into our heterologous pathway. As *AtoB* expression increases the available catalyst for the reaction of acetyl-coa into the mevalonate pathway, increases in *AtoB* expression increases flux to mevalonate and away from biomass generation over the course of the fermentation. We now include this correlation analysis in Supplementary Fig. 3c which shows a very high coefficient of determination (R^2) indicating a correlation between *atoB* expression and final mevalonate titer.

We added reference to this finding in the main text as follows: “Interestingly, the final titers achieved for each strain were highly correlated with the expression level of *atoB* (Supplementary Fig. 3c), the first dedicated step in the mevalonate pathway. It stands to reason that the expression level of *atoB* may dictate the metabolic flux, which is diverted from biomass accumulation as increased expression could increase flux into the mevalonate pathway.”

6. If e-formate and formate are chemically the same compound, then obtaining the same results is self-evident. If e-formate has any potential inhibitory effects on growth or production, please provide a description of this.

This is true, and we were pleased to see that the e-formate containing catholyte solution could be directly used, without purification, as a carbon and energy source for bioproduction. This is in part due to our choice of potassium carbonate catholyte solution which showed good biocompatibility and precluded the necessity for downstream purification.

To highlight these results the section on e-formate has been updated with the following sentence: “The catholyte solution showed no significant inhibitory effects on growth (Fig. 4b) and did not necessitate purification before use, simplifying its use as a carbon and energy source. “

7. The motivation behind using lignin as a substrate is unclear. While it does contain other organic acids, this study primarily focuses on formate, making the inclusion of lignin seem somewhat misaligned with the study’s main focus.

Thank you for drawing our attention to a lack of clarity in this section. We have now revised this section to highlight its importance in the context of formatrophic bioproduction and a formate bioeconomy. The novel lignin oxidation strategy generates formate as the most abundant product and our strain is able to utilize this formate while wild-type *E. coli* (and other major industrial hosts) cannot. Most importantly this lignin oxidation strategy is able to generate formate from a waste resource (lignin), which has a very long history of attempts to valorize. This may solve a very important problem for the formate bioeconomy which is that generating formate from electricity is too costly to be widely implemented for commodity chemicals. Hence, we think inclusion of our results

introducing this novel lignin oxidation strategy allows us to present a broad and encompassing argument for how formatotrophic *E. coli* like K4Me2 may be used in an industrial context for a formate bioeconomy.

Our choice of *Nature Communications*, a journal read by a wide and general audience, reflected our desire to craft a manuscript that would generate excitement about the concept of sustainable bio-economies from multiple perspectives. We aimed to encompass not only electricity-derived feedstocks but also lignin-derived feedstocks—an expansive and active field of research in its own right.

To better convey this message the text has been edited to:

1. **Highlight the importance of developing and exploring less costly means of generating formate than e-formate.**

“Despite its potential for sustainable, low-emission chemical production, bioproduction from e-formate or other electrochemical feedstocks is predicted to be relatively costly, limiting its near-term utility in the generation of low-cost, high-volume commodity chemicals.”

“As an alternative to costly e-formate, organic waste sources could be considered as a sustainable and inexpensive source of formate.”

2. **Reflect the advantages of our specific method of oxidative lignin depolymerization over existing methods.**

“Compared to other methods of lignin decomposition this oxidative method requires substantially less energy input as it is catalyzed at much lower temperatures (140 °C compared to 250-800 °C).⁶¹⁻⁶³ Additionally this oxidative route produces soluble, bioavailable acids in contrast to other methods, which produce hydrophobic lignin monomers or synthesis gas.^{61,64,65”}

3. **State directly the importance of utilizing a formatotrophic host which can take advantage of not only the widely metabolizable acids but also the relatively high amounts of formic acid present.**

“However, the formate (which is the most abundant product and constitutes 37% of the total bioavailable carbon identified in the mixture) cannot natively be used as carbon source by common industrial hosts and therefore would be wasted or else accumulate to toxic concentrations in an industrial fermentation.”

8. This study appears to address the issue of low FDH activity by increasing FDH expression levels. However, this does not necessarily solve the fundamental issue of low specific activity. Additionally, despite the increased expression levels, it is unclear why the flux is directed more toward product formation rather than growth. A discussion on this point would be beneficial.

Thank you for addressing this potential point of confusion. The text has been edited to make clearer that this study managed to achieve faster formatotrophic growth by fine-tuned, low levels of cnFDH expression which has a fast catalytic rate ($k_{\text{cat}} > 100 \text{ s}^{-1}$), about 20 times lower than previously needed high levels of psFDH expression (slower k_{cat} of $\sim 10 \text{ s}^{-1}$) (Kim, 2020 and Kim, 2023).

We have also added a supplementary figure to clarify our hypothesis explaining why the flux is directed more toward product formation rather than growth in the cnFDH bearing strains. The decreased allocation of proteome to FDH allows for higher expression of the mevalonate pathway in cnFDH strains, which we show in the newly added Supplementary Fig. 3c showing that the expression of the enzyme catalyzing the first step of the mevalonate pathway is highly correlated to the observed titer of mevalonate achieved.

We hope that the updated manuscript better conveys these hypotheses and recenters the key findings of our study, thank you for bringing this to our attention.